

# Defining the analytical complexity of decision problems under uncertainty based on their pivotal properties

Alexander Gutfraind[1,2,3]

[1] Epidemiology and Biostatistics, University of Illinois at Chicago, Chicago, Illinois, United States of America
[2] Department of Medicine, Loyola University of Chicago, Maywood, Illinois, United States
[3] Amazon Web Services, Herndon, Virginia, United States

## ABSTRACT

**Background:** Uncertainty poses a pervasive challenge in decision analysis and risk management. When the problem is poorly understood, probabilistic estimation exhibits high variability and bias. Analysts then utilize various strategies to find satisficing solutions, and these strategies can sometimes adequately address even highly complex problems. Previous literature proposed a hierarchy of uncertainty, but did not develop a quantitative score of analytical complexity.
**Methods:** In order to develop such a score, this study reviewed over 90 strategies to cope with uncertainty, including methods utilized by expert decision-makers such as engineers, military planners and others.
**Results:** It found that many decision problems have pivotal properties that enable their solution despite uncertainty, including small action space, reversibility and others. The analytical complexity score of a problem could then be defined based on the availability of these properties.

# INTRODUCTION

Solving most decision or risk management problems requires addressing uncertainty in all its forms (*Schultz et al., 2010*; *Yoe, 2019*). Although uncertainty could be modeled using probability theory (*Cox, 1946*; *Savage, 1951*; *Hacking, 2006*) probabilistic methods have come under criticism on theoretical and practical grounds since at least the 1920s in the work of *Knight (1921)* and *Keynes (1921b)*. To the critics the issues are: (1) It is impractical to build a forecasting model for the many outcomes of interest in the complex problems found in engineering, finance and other fields (*Sniedovich, 2012*; *Kay & King, 2020*); (2) Some problems are often too ambiguous or under-determined to be quantified using a model (*Regan, Colyvan & Burgman, 2002*; *Colyvan, 2008*); and (3) If a model could be built, the parameter estimates would be too variable or biased (*Gerrard & Tsanakas, 2011*; *Kay & King, 2020*) and in particular would involve low-probability, high-consequence events that are affected by high uncertainty (*Waller, 2013*) (for responses see *e.g.* (*Friedman, 1976*; *Aven, 2013*; *Waller, 2013*)).

Corresponding author
Alexander Gutfraind,
agutfraind.research@gmail.com

Various terms have been offered in the literature to describe this analytical complexity due to uncertainty including "Knightian uncertainty", "Hard uncertainty" (*Vercelli, 1998*), "Unknown Unknowns" (*Rumsfeld, 2011*), "Black Swans" (*Taleb, 2007*) and "Radical uncertainty" (*Kay & King, 2020*) with broadly similar meanings. However, these terms are quite difficult to operationalize and the best-known cases of unknown unknowns are disputed (*Aven, 2013*; *Wucker, 2016*; *Ale, Hartford & Slater, 2020*).

This article's aim is to introduce the framework of *pivotal properties* and show that these properties enable a definition of the analytical complexity of decision problems. Pivotal properties are linked to solution strategies so that once pivotal properties are identified the problem could be solved (or at least, satisficed) despite the uncertainty.

The field of forming a taxonomy of uncertainty has an extensive literature, notably in foundational categories such as *Knight*'s *(1921)* distinction, Hacking's identifying aleatory and epistemic uncertainty (*Hacking, 2006*), the distinction of uncertainty from ambiguity (*Colyvan, 2008*) and of modeling uncertainty (*French, 2023*). Recently, there have been domain-specific taxonomies of uncertainty (*Regan, Colyvan & Burgman, 2002*; *Ristovski et al., 2014*). *Lo & Mueller (2010)* identify five levels of progressively higher uncertainty based on system complexity. Concurrently, the computer science community developed a rich theory of computational complexity (*Arora & Barak, 2009*) but this theory only considers problems that could be defined in the strict formal context of Turing machines, rather than the complex socio-technical context of the decisions considered here (*i.e.* decisions by engineers, planners, decision consultants and others).

The project is positioned in Herbert Simon's tradition of satisficing (*Simon, 1955*). Various authors in the behavioral economics literature identified decision heuristics used by everyday consumers and by experts (*Gollier & Treich, 2003*; *Katsikopoulos & Gigerenzer, 2008*; *Binmore, Stewart & Voorhoeve, 2012*; *Todinov, 2015*; *Gutfraind, 2023*). More broadly, the tradition of naturalistic decision making found that decision makers often approach problems and evaluate solutions differently than what classic decision theory prescribes (*Klein, 2008*; *Tuckett et al., 2015*). If these powerful strategies could be utilized to address hard problems, then we must ask what problems remain hard. In the more classical probabilistic tradition the hardness of decision problems is primarily a function of forecasting difficulty (*e.g.* probabilistic estimation) or dimensionality (*e.g.* the number of parameters and cost of finding optimal solutions). However, if satisficing heuristics could adequately solve many socio-technical problems (of course, with no guarantee of optimality), a new approach to defining complexity is needed, and it is constructed here.

## METHODS

### Two contrastive examples

To motivate the development of our framework, we will review two simplified examples of uncertainty from the large list of scenarios underlying this study. The first example is the problem originally considered by both Knight and Keynes of entrepreneurial investment (*Keynes, 1921a*; *Knight, 1921*). An investor is seeking to allocate funds among firms and the goal is to maximize the long-term value of the total investment portfolio. Suppose the

business models or technologies are fundamentally new and so there is little quantitative basis for forecasting profitability any determinants of profitability, *e.g.* whether the market will see value in the products, whether the firms will be well-managed, or whether the economic and financial conditions will be favorable over the next 10 years. General benchmarks and trends are of course available for such investments, but they suffer from high dispersion and provide little specific guidance for choosing investments. For a second example problem, consider the resource allocation problem of a manager of a small health department. Novel pandemics are rare and hard to predict (*Marani et al., 2021*) creating a challenging planning problem. For more concreteness, suppose both decision-makers have wide discretion over a budget of 10 million dollars that must be allocated over 2 years and both must choose between several firms or several projects in the health department, respectively.

Contrasting the disparate problems in these scenarios suggests two properties of decision problems that we will call *pivotal*. While investment decisions are nearly entirely based on a single objective (future capital value), pandemic decisions must weigh multiple objectives (*e.g.* mortality, morbidity, economic impacts) and multiple stakeholders (*e.g.* different demographic or political groups). Secondly, risk in financial decisions can often be transferred to another party–for example, by pooling resources with another investor. In pandemic preparedness the major risks to health cannot be transferred in a similar sense.

Yet the two scenarios are similar in some surprising ways. Uncertainty in both scenarios, one might argue, could potentially be reduced through the acquisition of knowledge, *i.e.* the problems are Learnable: *e.g.* research in the technology or hiring expert trainers to educate the staff could fruitfully inform the decisions. This property should not be taken for granted because in some situations we lack a clear guide as to where our knowledge gaps are, or we might lack the ability to carry out a study since the system is too complex, inaccessible with current methods, or not subject to experimentation, as occurs in complex systems in fields such as public policy. Indeed, both problems are also what might be called well-understood: there is a body of scientific and practical knowledge around both investment and infectious diseases.

It is also relatively easy, one might argue, to find a feasible solution for the allocation of resources, *i.e.* both problems are easily Satisficeable. By contrast some problems are much harder to satisfice (*e.g.* building a rocket that will reach orbit or guarantee at least 10% annual return). Additionally, in what could be called indexability, the outcomes in both situations could be expressed adequately using random variables (*e.g.* the market value of the investment, the number of infections and a few others). Indexability should be distinguished from quantification (*e.g.* the ability to model or predict): the price of a company's stock indexes the value of the investment but does not eliminate the problem of economic forecasting or black swan events. Indexability enables estimation of volatility and an interactive trial-and-error approach to decisions until the outcome is improved (even if the system is difficult to study or model).

Lastly, the decision-makers in both problems are not mere passive observers to the phenomenon but are like engineers and have a large ability to configure or reconfigure the

solution. They can hire staff, design procedures, prepare contingency plans and so on. This Configurability property is used extensively in engineering, management and other settings to make physical systems or organizations robust to uncertainty.

Generalizing from these observations, we will attempt now to identify a comprehensive list of pivotal properties. With these properties we will define analytical complexity of a problem, *i.e.*, how hard it would be for a decision-maker or analyst to find a satisficing answer to the problem?

## Setting

Generalizing from the above examples, we are interested in characterizing a broad range of problems affected by a high degree of uncertainty and identifying properties that could aid in their classification according to a scale of analytical complexity. The problems of interest are all problems studied in decision theory and risk analysis: individual choices and large-scale societal problems; decisions with stakes both low and high. The goal of the problem might be optimization of an outcome subject to uncertainty or reduction of risk where the risk is difficult to quantify.

To be a little more precise, we will utilize the states-actions-outcomes framework (*Borgonovo et al., 2018*). The decision-maker (or makers) face an uncertain current state of the world, $s \in S$ and have to choose between actions $a \in F$ from a set that might be infinite. An action is a function that maps states of the world to outcomes (also called objectives): $G_a(s) \in O$. Our decision-makers are often unaware of the possible actions and outcomes (*Steele & Orri Stefansson, 2021*), unlike in classical settings. The action set might be very small (*e.g.* choosing between two options) or very large (as happens in problems of system design). In most cases we will think of the action as occurring at a single point in space and time but in certain situations we face multi-stage problems as well as an infinite sequence of decisions. In most problems, uncertainty about the state of the world $s$ could be decomposed into two factors: $Y$-the state of the system managed by the decision-maker (a vector-valued random variable) and $X$-context external to the system (another vector-valued random vector). The desired knowledge might be about these variable's present or future state, or certain performance outcomes that are functions of these variables. Many of the problems involve limiting the undesirable effects of $X$ on $Y$. Lastly, when speaking of risk we are referring to the qualitative notion of possibility of unfortunate occurrence (*Aven et al., 2018*).

Solving a problem here usually means finding a satisficing solution in the tradition of Herbert Simon (*Simon, 1955*; *Artinger, Gigerenzer & Jacobs, 2022*). Only in exceptional cases would the solution be optimal in any sense of the word. The *pivotal property* of a problem is defined as that property of the problem (*i.e.*, context, system or outcome) that could assist a decision-maker to solve the problem.

## Data source

The basis for this work is two lines of research. The first, coming from behavioral economics, is a list of decision heuristics being used in diverse fields (*Kahneman, Slovic & Tversky, 1982*; *Gigerenzer & Goldstein, 1996*; *Artinger, Gigerenzer & Jacobs, 2022*). The
second, a catalog built by this author and expanding the work of *Todinov, (2015)* is a list of strategies used by practitioners in fields such as engineering, management and others for decision making and risk mitigation (*Gutfraind, 2023*). Combining these lines of research resulted in a list of over 90 diverse strategies ranging from multi-layered defense popular in engineering to event tree analysis used in decision analysis[1]. The strategies were sourced from articles, monographs and interviews. The author also contributed examples from personal experience, primarily in engineering, software development, consulting and counter-terrorism. Important examples came from monographs across diverse fields: military science (*Freedman, 2015*), project management (*Flyvbjerg & Gardner, 2023*), safety and reliability engineering (*Stamatis, 2014*), software devops and security (*Howard, Curzi & Gantenbein, 2022*) and biology (*Ellner & Guckenheimer, 2011*). Additionally, the author hired anonymous assistants through the site fancyhands.com. They were tasked to list examples of problems they solved in their daily lives (all details anonymized). The solution strategies were then included in the list, if they presented a general-purpose technique for uncertainty.

[1] List of the RDOT strategies for uncertainty management is archived at https://doi.org/10.5281/zenodo.8350550

### Identification of properties

Based on this knowledge base the next step was to find pivotal properties. The process was entirely qualitative and several prompting techniques were used. The first approach was to review each of the strategies in the knowledge base and to ask which properties must the setting have in order for the technique to be applicable to a new domain. It was generally quite clear what some of the properties must be. The second approach was to contrast two problems and ask, what are essential differences in the problem settings. The third technique was to ask, for a given problem, what parameters could make the problem easier or harder. The fourth step was to review the emerging list of properties and consolidate closely related properties into a group with a common definition. Because each problem has many unique properties, a property was only included in the list below if, (1) it was logically necessary for applying a solution heuristic and, (2) it was shared by two or more problems or strategies coming from different application domains. Finally, the properties were put into clusters.

## RESULTS

### Pivotal properties

Analysis of the knowledge base produced 19 pivotal properties that are listed below in alphabetical order. The properties could be organized into four loose clusters based on what they relate to: the problem objectives, the action space, the role of time, and the nature of uncertainty. To clarify the meaning of each property, we list solution heuristics (*i.e.* strategies) that depend on it.

### Properties related to the problem objectives

#### Few objectives or stakeholders

Definition: Only one or several objectives are important or, equivalently, the decision needs to satisfy only a small number of stakeholders or classes of stakeholders. The

significance of this property is discussed in *Arrow (1950)*, *Papageorgiou, Eres & Scanlan (2016)*.

Solution methods: Mathematical optimization of total outcome, decision by polling or delegation.

### Indexable outcomes (Mankelwicz, 2010)

Definition: Outcomes could be measured or expressed numerically.

Solution methods: Statistical modeling, Reinforcement learning (*Sutton & Barto, 2018*).

### Reversible decision (Verbruggen, 2013)

Definition: The decision/action could be revoked or substituted in the future with relatively low costs.

Solution methods: Trial-and-error, Wait and see.

### Satisficeable decision (Simon, 1955)

Definition: Problems where feasible solutions are easily found and/or where many of the solutions are likely acceptable to the stakeholders.

Solution methods: Heuristic solution (*Gigerenzer & Goldstein, 1996*), Managerial assumption, Use the default action.

## Properties related to the action space

### Collective or organizational action

Definition: The actions are planned or implemented by multiple agents or an organization.

Solution methods: knowledge dissemination, coordinated response, delegation (*Ben-Shalom & Shamir, 2011*).

### Configurable system

Definition: The action space includes designing or improving a system under decision makers' control.

Solution methods: Multi-layered defense (*Nunes-Vaz, Lord & Ciuk, 2011*), resilient design (*Ganin et al., 2016*), incident response unit, prototype-driven development.

### Simplifiable solutions

Definition: The ability to use a simple version of the solution to learn and optimize solutions and to reduce the risks of implementing a full-scale solution.

Solution methods: Prototype-driven development (*Thomke & Bell, 2001*), mathematical modeling, virtual simulation.

### Small action space

Definition: The set of possible actions is relatively small, allowing detailed analysis.

Solution methods: Exhaustive analysis of all actions, game-theoretic analysis (*Myerson, 2013*).

### Transferable risk

Definition: The risk could be transferred substantially to another party.

Solution methods: Risk contracts (*Gárdos, 2022*), innovative insurance & financial instruments (*Tufano, 2003*).

## Temporal properties
### Repeating resource decisions
Definition: The problem involves repeating decisions involving resources over sequences of time.
Solution methods: rK strategy (*Travis et al., 2023*), portfolio rebalancing (*Dichtl, Drobetz & Wambach, 2016*).

### Sequentially interactable system
Definition: The ability to perform actions on the system and observe outcomes, investigate it gradually learning the system and to optimize actions.
Solution methods: Reinforcement learning (*Sutton & Barto, 2018*), stochastic search, trial-and-error, incident investigation (*Stemn et al., 2018*).

### Temporal enablement
Definition: Situation of sequential decisions where progression of time can enable new capabilities, options or pivotal properties.
Solution methods: Fundamental research, investment in enablement of new strategies, evolvable design (*Urken, "Buck" Nimz & Schuck, 2012*), Real options (*Brach, 2003*).

## Properties of the uncertainty
### Containable hazard
Definition: The event could be contained in space, time or other dimensions, as opposed to necessarily affecting the entire system, outcomes or objectives.
Solution methods: Minimize area or time of impact (*Manadhata & Wing, 2011*), deflect or delay hazard, automatic containment system (*Kopetz, 2003*).

### Delayed onset or drawn-out impact
Definition: Adverse events are anticipated to occur after a relatively long period of time, or to have gradual impact.
Solution methods: Fundamental research, expansive analysis, contingency planning (*Clark, 2010*).

### Detectable hazard
Definition: Undesired events could be recognized before they occur or soon thereafter.
Solution methods: Early warning system (*Quansah, Engel & Rochon, 2010*), automatic containment system (*Kopetz, 2003*), event forensics and attribution.

### Known adversary
Definition: The decision must consider a rational agent that acts to improve their objectives and possibly reduce ours, but we have knowledge about their objectives and capabilities.

Solution methods: Game-theoretic analysis (*Myerson, 2013*), misdirection/Deception, accelerate adaptation (*Hoffman, 2021*).

### Learnable

Definition: Additional research, data gathering and analysis of the problem are possible.
Solution methods: Basic research, gathering data, meta-learning of unknowns, knowledge dissemination.

### Small event space

Definition: The set of possible events is relatively small.
Solution methods: Contingency planning (*Clark, 2010*), fault tree analysis (*Ruijters & Stoelinga, 2015*).

### Well-understood

Definition: A body of knowledge exists to describe the phenomenon and problem such as scientific understanding, data or human experience.
Solution methods: System models (mathematical, computational, predictive), maximization of expected utility, decision templates, expert elicitation and judgment (*Meyer & Booker, 2001*).

For each property Fig. 1 lists the total number of dependent solution strategies. Because the assignment is a matter of judgment, the count should be viewed as suggestive of the importance of the property.

### Analytical complexity score of decision problems

Using the pivotal properties we can develop a system to assess the analytical complexity of decision problems. Recall that a property of a problem is called pivotal if it could assist a decision-maker to solve the problem. Therefore, a problem's complexity is linked to the availability of pivotal properties, and an overall complexity score could be developed.

To define this score suppose that for problem $z$ the $n$ pivotal properties are described by the vector $Q(z) \in \Re^n$. Thus, $Q_i(z)$ gives a score for property $i$ according to its own scale (*e.g.* the number of stakeholders, cost of reversing a decision *etc*). Then the complexity score could be function of the form: $H(z) = h(Q(z))$ where the function $h : \Re^n \to [0, 1]$ is non-increasing in the arguments.

An empirical approach to computing complexity would be to statistically estimate the complexity from a dataset of problems. To implement this approach we would need to build a dataset of problems from different domains and for each problem measure the pivotal properties and its complexity expressed in terms of effort[2] to make a decision. Then we could estimate the function $h$ by applying logistic regression (or generally a statistical estimation method) on the vector $Q(z)$.

In the absence of such data, we could apply a functional form that would give us a simple albeit arbitrary measure of complexity, as follows. Suppose $R_i(z) \in [0, 1]$ gives the

---

[2] This could be quantified in terms of hours of work or dollars in consulting fees, adjusted for overall project cost. The exercise is by no means trivial.

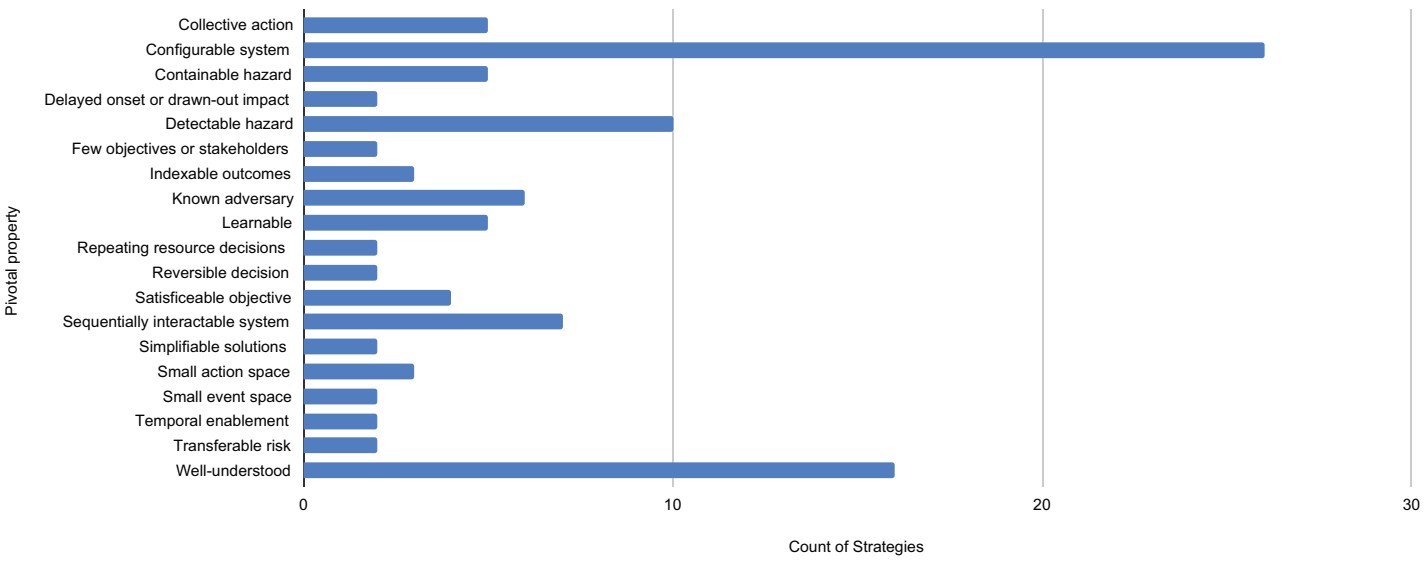

**Figure 1** **Pivotal properties of decision problems and the number of solution strategies that depend on them.** The 19 pivotal properties enable from 2 to 26 strategies.

extent to which the property $i \in \{1, 2, \ldots n\}$ resolves the problem (resolving will be defined below). Analytical complexity of a problem could then be defined as the product:

$$H(z) = \prod_{i \in \{1, \ldots, n\}} (1 - R_i(z)). \tag{1}$$

In general, we will need a separate $R_i$ for each property. Using the product function in Eq. (1) reflects the effect that the complexity of the entire decision problem can vanish if $R_i = 1$ for any $i$. For example, any decision problem is analytically trivial if there is only one possible action. Additionally, the product function guarantees that finding additional pivotal properties in other problems would not affect $H(z)$ since they do not assist in resolving $z$.

In the case of sequential decisions or decisions where only some of the pivotal properties could be used, we must need to compose the $R_i(q)$ values using a different function (not a product). The simplest version of this definition is to make $R_i \in \{0, c\}$ for some positive constant $c \in (0, 1]$, *i.e.*, just indicate the absence or presence of the pivotal property in problem $z$. Then the complexity of the problem simplifies to $H(z) = (1 - c)^{I(z)}$, where $I(z)$ is the number of pivotal properties of $z$.

## DISCUSSION

This study is a reexamination of complexity in decision theory and risk. It is inspired by the paradoxical situation where, on the one hand, virtually all problems are profoundly affected by uncertainty, and yet practitioners seem to have a large arsenal of solution strategies. The paradox is resolved by noting that problems have properties that provide a

backdoor to finding good solutions. These properties allow us to quantify the analytical complexity of problems.

At a practical level, it can help estimate at a high level the analytical resources that would be needed to tackle a given decision problem, such as time, computational resources and data collection. The complexity score of each practical problem might be somewhat affected by unique factors that are unique to it, and the score is best viewed as a high-altitude picture of analytical complexity similar to Reference Class Forecasting (*Flyvbjerg, 2006*). The design of the scores using the regression or product methods ensures that if additional properties are found, they will refine the scores rather than overturn them. Similar to how certain chaotic *i.e.* unpredictable physical systems can have their parameters computed precisely (*Peng & Li, 2022*), the definition can give a usable measure of complexity even for problems that are largely difficult or suffer from radical uncertainty.

The input values of the complexity formula are relatively objective statements of whether particular properties hold in the problem. However, because the values of $Q_i$ and $R_i$ are hard to measure, the Eq. (1) is best applied for comparing two related problems. Some of the pivotal properties are analyst-dependent (especially, whether a problem is "well-understood") and non-public information could assist some decision-makers in solving the problem (*Savage, 1951*; *Ramsey, 2013*; *Bier & Lin, 2013*; *French, 2022*).

The advantage of the proposed analytical complexity is that it helps us move away from dichotomous and disputed terms such as the "black swan" or "unknown unknown". While many problems are affected by profound uncertainty, they might be easy from a decision-theoretic point of view. For example, long-term investment decisions are classical examples of Knightian uncertainty, but they are easy if the number of choices is small. Many investment advisors merely propose that clients choose whether to invest 60% of their portfolio in an index of stocks and the rest in bonds, or *vice versa*. In our analysis of the empirical dataset described in Methods, in every problem we could always find some pivotal properties and mitigation strategies.

One may hypothesize that the majority of problems that are affected by large uncertainty (and are hard to address with expected utility methods) are nevertheless solvable. A seemingly unpredictable planetary disaster, such as an asteroid impact (*Reinhardt et al., 2016*) has many pivotal properties: humanity has a limited set of objectives (foremost, survival), the action and event space is relatively small, the event is detectable and learnable. This is not to deny that there exist situations, the hardest nuts, that have few pivotal properties and consequently cannot be attacked with any of the strategies. We can imagine such a scenario by negating the pivotal properties; *e.g.*, poorly understood, hard to learn, non-stationary, not detectable in advance, lack of system design or control over response. Good examples of hard problems are those involving complex adaptive systems that are connected to human society in complex ways. Examples, therefore, include environmental and economic problems-the well-known area of "wicked" planning problems (*Rittel & Webber, 2018*).

There are both empirical and conceptual limitations to this study. First, analytical complexity measures how hard it is to find a solution but not the complexity of its implementation in terms of resources, commitment and time. It best serves consultants

and other users who are responsible for designing and planning rather than executing the solution. Second, the underlying knowledge base of strategies is incomplete-using just around a hundred strategies, with particularly limited coverage of certain policy and environmental problems too far outside the author's expertise. The knowledge base also is focused on solving in the presence of uncertainty and may omit other complexifying factors for decision making.

Lastly, the process of identifying pivotal properties to a strategy requires a qualitative evaluation and expert judgment. The results are adequate for constructing a list of the pivotal properties but future work should ideally revisit this exercise using a fully documented multi-expert process (*O'Brien et al., 2014*; *Baker, Hodges & Wilson, 2018*). Such a process is expected to refine and expand the pivotal properties named above.

## CONCLUSIONS

This study aims to better address the challenge of uncertainty within decision theory. Drawing on the toolkit developed by expert practitioners, it found that decision problems have pivotal properties and offered a new framework to quantify analytical complexity. The pervasive challenge of uncertainty suggests that additional methodological advances could bring practical value to decision making and risk management.

## ACKNOWLEDGEMENTS

Discussions with Professors Vicki Bier and Michael Genkin and several colleagues helped refine this study. I thank the anonymous reviewers for suggestions that greatly enhanced this paper.

### Funding

The project was sponsored by NIH grant R01-AI158666. There was no additional external funding received for this study. The funders had no role in study design, data collection and analysis, decision to publish, or preparation of the manuscript.

### Grant Disclosures

The following grant information was disclosed by the authors:
NIH grant R01-AI158666.

### Competing Interests

The author is an employee of Amazon Web Services, however the study is not related to his employment and any positions in this study are his own.

### Author Contributions

- Alexander Gutfraind conceived and designed the experiments, performed the experiments, analyzed the data, performed the computation work, prepared figures and/or tables, authored or reviewed drafts of the article, and approved the final draft.

## Data Availability

The list of the RDOT strategies for uncertainty management is available at Zenodo: Sasha Gutfraind. (2023). sashagutfraind/uncertainty_strategies: v1 (Version v1). Zenodo. https://doi.org/10.5281/zenodo.8350550.

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
