# Peer review of "Defining the analytical complexity of decision problems under uncertainty based on their pivotal properties"

_PeerJ Computer Science, doi:10.7717/peerj-cs.2195_

## Round 0.1 · original submission · Major Revisions

Dear Dr. Alexander Gutfraind,

Subject: Decision on Manuscript ID CS-2023:09:90493:0:1:REVIEW - "Defining the analytical complexity of decision problems under uncertainty based on their pivotal properties"

I am writing regarding your submission to PeerJ Computer Science. After careful review, we have concluded that your manuscript requires major revision before further consideration.

Your paper, which delves into decision-making under radical uncertainty, is recognized for its innovative approach and integration of real-world strategies. However, there are several key areas where revisions are necessary (please, see the detailed comments of the reviewers):

The reporting of data and analysis methods requires enhancement. More detailed and transparent reporting, as per standard qualitative research protocols, is essential.

There is a need for greater clarity in the definition of key terms such as 'risk', 'uncertainty', and 'complexity'. Consistent and clear definitions will strengthen your argument.

The mathematical models and their practical applicability in real-world scenarios need to be revisited. Addressing the limitations and potential improvements in these models is crucial.

Certain claims in the manuscript, particularly those around forecasting capabilities, require more robust justification or a more modest approach.

The selection of research articles and problem settings could benefit from a broader scope, especially considering the complexities of wicked problems in environmental decision-making.

The manuscript should also consider individual differences in decision-making, acknowledging the varying expertise and perceptions of decision-makers.

We believe that your manuscript has the potential to make a significant contribution to the field. However, addressing the above points can improve your work. We look forward to receiving your revised manuscript.

Please feel free to contact us for any clarifications or further discussion.

Sincerely,

José Manuel Galán
Academic Editor, PeerJ Computer Science

·

Basic reporting

This paper addresses an important topic, approaches to effective action in the face of radical uncertainty. The central approach of the paper, drawing from accounts of decision strategies used by expert decision-makers is potentially valuable. Intriguingly the author, essentially draws on qualitative research to motivate a taxonomy of approaches to acting in the face of radical uncertainty. This forms the basis of identifying pivotal properties of problems that may be used to ‘solve’ decision problems and a mathematical approach to assess the ‘analytical complexity’ of problems.

It is particularly commendable that the author has sought to draw on a wide range of ‘real world’ strategies for addressing conditions of radical uncertainty and this is in the best traditions of naturalistic decision-making research (see e.g. Klein, G. (2008). Naturalistic decision making. Human factors, 50(3), 456-460.)
The paper is written in clear English and develops a line of argument. However, I do have some major concerns about the basic reporting of the study and by implication the inadequate account of study design and the adequacy of argumentation.

1. The paper draws on qualitative accounts of expert decision-making and action to address problems characterized by radical uncertainty. However, the underlying data are not reported nor are methods for analyzing the data. What reporting of data and methods there is relies on reference to a pre-publication paper (Gutfraind, 2023), which again offers an inadequate account of data and methods. A fundamental principle of research writing is that the reader should have sufficient information to judge whether they would draw the same conclusions from the data as the author. There are a number of standards available for the reporting of qualitative research, which could be used. See for example: O’Brien, Bridget C. PhD; Harris, Ilene B. PhD; Beckman, Thomas J. MD; Reed, Darcy A. MD, MPH; Cook, David A. MD, MHPE. Standards for Reporting Qualitative Research: A Synthesis of Recommendations. Academic Medicine 89(9):p 1245-1251, September 2014. | DOI: 10.1097/ACM.0000000000000388

2. This lack of information about the quality of the data you have collected is of especial concern since the (large) literature on expertise suggests that when asked to describe what they do experts often give erroneous accounts or miss important information. For this reason much expertise research relies on observational studies and ‘think aloud’ protocols. See e.g. Ericsson, K. A. (2018). 12 Capturing Expert Thought with Protocol Analysis: Concurrent Verbalizations of Thinking during Experts’ Performance on Representative Tasks. The Cambridge handbook of expertise and expert performance, 192.

3. Given that the goals of the paper include mathematical formalisation of approaches to ‘solving’ problems under uncertainty, there are too many instances of inadequate definition of key terms. For example, you seem to use the term ‘risk’ in a number of different ways, including in contrast to uncertainty, and as a generic term for hazard. In economics, risk is often used as a term for a probability distribution function (including both ‘upside’ and ‘downside’). In much social science it is used as a term for hazards that may or may not occur (downside only). Similarly, although more care is taken in discussing competing definitions of uncertainty, your precise definition of the term as you will use it is left unclear. It is also common to distinguish carefully between complexity and uncertainty, yet you move between these two terms without sufficient clarity about the ways in which you define them. To give a final example, you discuss problem solution without sufficient clarity about what it might mean to solve a problem of action under uncertainty, or what might characterise the difference between a better and worse ‘solution’, or perhaps between a ‘good enough’ and a ‘not good enough’ solution.

4. You make too many grandiose claims without sufficient justification. For example, it seems odd to claim that ‘seemingly impossible forecasting problems such as forecasting sudden geopolitical events and financial crises, have been solved’ (lines 54/55). Tetlock’s point about judging forecasts is that reputation for forecasting often rests on predictions that some event will happen on an indeterminant timescale whereas the hard problem is forecasting within what timescale an event is likely to happen. (See eg. Tetlock, P. E. (2017). Expert political judgment. In Expert Political Judgment. Princeton University Press.) For example, most macro economists are fairly certain that there will be another global financial crisis but profoundly uncertain about likely timescale and nature. The very large majority of climate scientists agree that the climate will continue to heat and there is some degree of consensus on the global consequences. However, there is great uncertainty about the nature of weather changes in any specific region of the planet with forecasts for specific regions ranging from increases in drought to increases in flooding and from local cooling to local heating. And so on. Further, most work on effective foresight (again a large literature) emphasises the priority that should be given to exploration of multiple alternative futures over prediction. For example, Dator (2019b) has argued that ‘[n]othing in society, beyond the most trivial can be precisely predicted. … The future cannot be predicted, but alternative futures can and should be forecast’.

Experimental design

See basic reporting

Validity of the findings

There are a number of threats to the validity of the conclusions of this paper.

1. As noted above, inadequate reporting of data and analysis compromises the value of the taxonomy, that is developed from it, of pivotal properties and ‘solution’ strategies.

2. Whilst it is normal that mathematical representations necessarily simplify the world they represent and abstract from, it is important that they do not eliminate strongly salient problem elements. The ever-present danger is that achievement of mathematical tractability and generalizability is at the expense of adequate representation and hence usefulness in tackling real problems. In this regard the proposal for calculating analytical complexity seems likely to be beset by intractable problems of adequate representation in most real-world problems that involve radical uncertainty.

Take, for example, your suggestion of number of stakeholders as one property that might enter into a calculation of analytical complexity. There is a large literature on stakeholder analysis, with many more important characteristics of stakeholders than number. These include moral claim on an interest in outcomes, power to sanction or reward decision-makers whether directly (e.g. a regulator) or indirectly (e.g. via impact on an organisation’s reputation) and diversity of goals and perspectives among stakeholders. It is rare that any of these can be known with any precision or sometimes even meaningfully be translated into a scalar measure. It is likely that many of the important variables that might be used to construct such a measure raise similar problems. As Knight notes in the book you cite “The assumption that under the same circumstances the same things behave in the same ways thus raises the single question of how far and in what sense the universe is really made up of such "things" which preserve an unvarying identity (mode of behavior). It is manifest that the ordinary objects of experience do not fit this description closely, certainly not such "things" as men and animals and probably not even rocks and planets in the strict sense. “ This sense of Knightian Uncertainty is a significant problem for mathematical representation, most especially scoring.

You do go on to propose a simpler approach, such that properties are scored in the real interval [0,1] on the extent to which the property resolves the problem. Here I am, firstly, just puzzled about why you need a mathematical formalization since before attempting a calculation, you already know that the problem has a property that can resolve the problem. Overall, I am doubtful of the value your mathematical formalization currently adds beyond your taxonomy of pivotal properties and the strategies they relate to.

3. In your discussion, you sensibly note that “The best examples of hard problems are perhaps those involving complex adaptive systems that are connected to human society in complex ways.” However, much of the literature that you critique (e.g. Kay and King, Keynes, Knight etc.) considers exactly these kinds of problems as typical examples of radical uncertainty. Biological and social systems are typically complex open systems, in which there is relentless innovation in response to competition for resources. The outcome is that such systems are typically non-ergodic except on short time scales. One outcome is that models of such systems often work until they fail (often unexpectedly and in unexpected ways, as in the 2007/8 global financial crisis). At the very least, you should be more more modest about the potential impact of this work.

Additional comments

By this point, you may have the sense that I have formed an entirely negative view of the paper. That is not the case. I do believe the paper needs substantial redevelopment. However, I also believe that there is a potentially useful contribution here. In particular, I believe that with more effective discussion of the methodology adopted, to source and analyse the taxonomies you develop, that there could be a good contribution to understandings of how domain experts grapple with uncertainty with implications for improving practice. I take your approach to be something like thematic analysis ( See e.g. Nowell, L. S., Norris, J. M., White, D. E., & Moules, N. J. (2017). Thematic Analysis: Striving to Meet the Trustworthiness Criteria. International Journal of Qualitative Methods, 16(1). https://doi.org/10.1177/1609406917733847).

However, your discussion of pivotal properties and strategies would benefit from more detailed explanation of each element and some examples. I think the paper would also very much benefit from more careful discussion of what counts as a ‘solution’ in this space.

The construct of pivotal properties is useful and could be a genuine contribution. I am tempted to advise dropping the mathematical formalization of these insights, but would suggest that at least you need to pay rather more attention to how this could add value and the practical difficulties of calculating the kinds of scores you propose without throwing the baby out with the bathwater. In doing this it would be helpful to work, in detail, though an example, perhaps with the aid of a domain expert for that example.

Reviewer 2 ·

Basic reporting

The article has mostly been written with a clear English. However, some of the sentenced are hard to understood for non-fluent English speakers even though they are written professionally. Article include sufficient background and context with enough literature references. Structure of the article is fine.

Experimental design

The research question is well defined and important to the research field. However, the selection of only 90 research articles seems to be rather small taking account the broad spectrum of decision-making problems. Furthermore, it seems that most of the problems are from fields with narrow problems settings are not wicked like most of the environmental decision-making situation. This should be clarified (even though wicked problems are mentioned in a chapter Discussion)

Validity of the findings

The results that decision problems have pivotal properties is certainly true and valid and observation that they usually are even mixed seems to be valid also. In fact I argue that for wicked problems this is even pivotal property.

The derived equation 1 seems to be adequate to describe the complexity for simple (sequential) decision-making problems. For some decision-making processes situation might however be such that this kind of linear approach is not sufficient i.e, in a situation were founded pivotal properties are exclusive. I mean that decision making process is a branching one. (in that case factors 1-Qi should be summed?)

Additional comments

Lines 324: The costs are not defined clearly enough, for different problems cost can mean totally different things.

The scale of Qi(z) change. In a line 319-320 it has different scale than in the line 330


In line 347 auhor argue that: “analytical complexity will be the same for all equally-informed decision-makers.” I have to say that I disagree. Eventhought every decision maker has given the same data and knowledge of the all processes there are always small differences between how they actually see things and therefore this complexity indicator get different values. These differences arise from the expertise of the decision makers. In a large and complex problems, one cannot be expert of everything and thus handle the information differently.

---

## Round 0.2 · Minor Revisions

The paper has been improved as mentioned by the reviewers. I have decided to make minor revisions to give you the opportunity to include the suggestions pointed out by the reviewer if you consider doing so, but the work is acceptable for publication. Congratulations.

·

Basic reporting

Thank you for clarifying the nature of the data on which you draw. I believe it is now clear that you derive examples of strategies for addressing uncertainty from prior literature on behavioural economics, heuristics and decision-making, and naturalistic decision-making. You also say "Additionally, the author and several paid assistants collected examples of problems from various domains and recorded strategies/heuristics applied for solving them.". You should provide further detail on how you did this and the domains and sources of information used.

Experimental design

NA

Validity of the findings

In my judgement, this paper adds usefully to conversations about responding to deep or radical uncertainty in real world settings. The core contribution is to develop a taxonomy of pivotal properties which may support the use of strategies to enable action in the face of uncertainty.

There are three main ways in which I believe the paper could be usefully improved. First, you need to take great care to adequately define the key terms you use (especially where usage varies in other literature). For example, you seem at time to conflate complexity and uncertainty. Whilst uncertainty may arise from the emergent properties of complex systems (especially complex open systems), they are entirely different phenomena and you should take great care to clarify what you mean by each (it is also common in systems theory to make a clear distinction between complex systems and complicated systems). In this paper you are working at the intersection of disciplines, so you cannot assume a common understanding of terms.

Second, your taxonomy of pivotal properties and strategies they enable is very terse and lacks adequate exposition or explanation. Since the taxonomy is fairly lengthy, it will be hard to expand this a great deal, but in many cases you could reference prior literature that describes the property or strategy. In other cases a couple of additional lines could add clarification - including why the strategy is appropriate.

Third, and perhaps most important, any theory should consider boundaries of application. You nod towards this when you say "This is not to deny that there might exist situations, the hardest nuts, that have few pivotal properties and consequently cannot be managed with any of the strategies." I think you can make a stronger claim than "might" (such situations clearly do exist and may even be the larger category.) You should go beyond this to attempt some characterisation of the kinds of situations that cannot be managed by the strategies you consider. This should go beyond the one paragraph example you give; but this might usefully be an expansion of it.

Additional comments

You may find the following recent article helpful in positioning your approach relative to current research -
Tuckett, D., Mandel, A., Mangalagiu, D., Abramson, A., Hinkel, J., Katsikopoulos, K., … Wilkinson, A. (2015). Uncertainty, Decision Science, and Policy Making: A Manifesto for a Research Agenda. Critical Review, 27(2), 213–242.https://doi-org/10.1080/08913811.2015.1037078

Reviewer 2 ·

Basic reporting

The revised version of the manucript has improved a lot when compared to original one.

Experimental design

No comment

Validity of the findings

No comment

Additional comments

Methodology is valid but I am still quite skeptic how useful it would be in a complex decission making situation.

---

## Round 0.3 · accepted · Accept

The current version of the paper is now acceptable for publication